# Specific Incremental Test for Aerobic Fitness in Trail Running: IncremenTrail

**DOI:** 10.3390/sports10110174

**Published:** 2022-11-09

**Authors:** Grégory Doucende, Maxime Chamoux, Thomas Defer, Clément Rissetto, Laurent Mourot, Johan Cassirame

**Affiliations:** 1Laboratoire Interdisciplinaire Performance Santé en Environnement de Montagne (LIPSEM), Université de Perpignan Via Domitia, UR-4604, 7 Avenue Pierre de Coubertin, 66120 Font Romeu, France; 2EA 3920-Prognostic Markers and Regulatory Factors of Cardiac and Vascular Diseases and Plateforme EPSI, University of Franche-Comté, 25480 Besançon, France; 3Laboratory Culture Sport Health and Society (C3S−UR 4660), Sport and Performance Department, University of Bourgogne Franche-Comté, 25000 Besançon, France; 4EA 7507, Laboratoire Performance, Santé, Métrologie, Société, 51100 Reims, France; 5Mtraining, R&D Division, 25480 Ecole Valentin, France

**Keywords:** trail running, testing, uphill, method, oxygen consumption, validation

## Abstract

Trail running (TR) is performed in a natural environment, including various ranges of slopes where maximal oxygen consumption is a major contributor to performance. The aim of this study is to investigate the validity of tests performed in uphill conditions named the “IncremenTrail” (IncT), based on the incremental ascending speed (AS) to evaluate trail runners’ cardiorespiratory parameters. IncT protocol included a constant gradient slope set at 25% during the whole test; the starting speed was 500 m·h^−1^ (25% slope and 2.06 km·h^−1^) and increased by 100 m·h^−1^ every minute (0.41 km·h^−1^). Twenty trail runner specialists performed the IncT and a supramaximal exercise bout to exhaustion with intensity set at 105% of maximal AS (Tlim). Oxygen consumption, breathing frequency, ventilation, respiratory exchange ratio (RER), and heart rate were continuously recorded during the exercises. The blood lactate concentration and rate of perceived exertion were collected at the end of the exercises. During the IncT test, 16 athletes (80%) reached a plateau of maximal oxygen uptake (65.5 ± 7.6 mL·kg^−1^·min^−1^), 19 athletes (95%) reached RER values over 1.10 (1.12 ± 0.02) and all the athletes achieved blood lactate concentration over 8.0 mmol·L^−1^ (17.1 ± 3.5 mmol·L^−1^) and a maximal heart rate ≥90% of the theoretical maximum (185 ± 11 bpm). Maximal values were not significantly different between IncT and Tlim. In addition, ventilatory thresholds could be determined for all runners with an associated AS. IncT provided a suitable protocol to evaluate trail runners’ cardiorespiratory limitations and allowed us to obtain specific intensities based on the ascending speed useful for training purposes in specific conditions.

## 1. Introduction

Trail running (TR) is a generic term used to describe off-road running practice in a natural environment, including mountains, forests, deserts, and many other terrains [1]. TR competitions are organized with a minimum of asphalted road (20–25% maximum) and cover a large range of distance from 10 to more than 100 km with various ranges of elevation potentially over 10,000 m (e.g., the Ultra-Trail du Mont Blanc—UTMB^®^, 170 km and over 10,000 m of elevation). In recent decades, TR has become extremely popular, and the number of participants and events has increased exponentially. Compared to level road events of the same length, TR includes uphill and downhill sections on various slopes, that are known to alter the energy cost of running and generate greater muscular damage [2,3,4]. Moreover, uphill sections induce specific running kinematics [5,6] necessary to elevate the centre of mass at each step [7,8], and consequently greatly increase the energy expenditure [9].

Many recent studies have pointed out that TR performance is largely related to maximal oxygen consumption capability [3,10,11,12,13], reinforcing the interest of $\dot{V}O_{2max}$ assessment for performance purposes. The effect of gradient slopes on $\dot{V}O_{2max}$ has been largely studied and results differ according to the slopes gradient, protocol, or the participants. Considering the TR runner specialist population, few studies have investigated the effect of slope gradient on $\dot{V}O_{2max}$ reached during an incremental running test. Balducci et al. did not observe the modification in $\dot{V}O_{2max}$ between level and slope conditions during an incremental running test on a treadmill at a 12.5 and a 25% gradient [14]. More recently, Schöffl et al. reported similar $\dot{V}O_{2max}$ for a trail runner specialist and a road runner when comparing outdoor tests on a natural slope at 16% and a treadmill running at a 1% gradient [15]. In addition, De Lucas et al. proposed a new test based on a slope gradient increase instead of a horizontal speed increase and did not point to any difference between the two typologies of test [16]. In contrast to these results, Scheer et al. reported significantly higher $\dot{V}O_{2max}$ in a specific trail test performed on a treadmill with an increase in speed and slope over the test (1 km·h^−1^ and 1% gradient slope per minute) [17]. In line with these results, Cassirame et al. investigated physiological variables at ventilatory thresholds and at exhaustion during incremental field tests at the levels 15%, 25% and 40% of gradient slope [18]. This study indicated that uphill running allows highly trained TR specialists ($\dot{V}O_{2max}$ > 65 mL·kg^−1^·min^−1^) to reach higher $\dot{V}O_{2max}$ than those running in level conditions. Nevertheless, this study also demonstrated that $\dot{V}O_{2max}$ is not significantly different between a 25 and a 40% slope gradient and this gradient allows runners to achieve physiological limits equally.

Based on this statement, it appears of interest to investigate and evaluate TR specialists in uphill conditions to obtain physiological limitations and adaptations in situations closer than those faced during TR practice. In addition, we noted that the range of slopes and field typology required more specific intensity markers to prescribe training intensities or calculate training load [19]. In their study, Cassirame et al. [18] indicated that ascending speed (AS) can be a relevant indicator of the intensity for slopes between a 25 and a 40% gradient. In this study, $\dot{V}O_{2}$ reached at exhaustion or ventilatory thresholds in gradient tests set at 25 and 40% did not differ. For several years, AS has been calculated with GPS and or barometer technology and TR specialists can use this variable in real time on their heart rate monitor and analyze it after training. This AS concept is also largely influenced by challenge on social networks where athletes highlighted their maximum elevation performed in time from 1 h to 24 h.

Traditionally, running performance capabilities including $\dot{V}O_{2max}$ assessment are performed in level conditions with existing or adjusted tests as “VamEval” or the “Université de Montréal track” tests [20], or with a minor slope gradient (1–2%) when performed in treadmill conditions [21]. Classic tests cannot be performed in a slopes condition without adjusting the speed protocol to ensure physiological exhaustion [22] and lead to maximal oxygen consumption instead of peak values [23,24]. Previously, Scheer et al. [17] presented a trail test based on both slope and speed increase. Unfortunately, a starting protocol with a slope gradient lower than 25% does not ensure linear progression of AS and a simultaneous increase in slope and speed induces an abrupt increase in energy expenditure [9,25] and, consequently, an AS determined during the test could not be used for training. For these reasons, a specific test named “IncremenTrail” (IncT) [26] has been developed in order to evaluate $\dot{V}O_{2max}$ and the AS. Contrary to classical tests, the IncT is based on an incremental AS instead of a horizontal speed with a constant slope set at a 25% gradient. This gradient was selected based on a previous study indicating that a maximal physiological response can be observed at 25% [18]. The IncT protocol starts at 500 m·h^−1^ of AS (25% slope and 2.06 km·h^−1^) and increases by 100 m·h^−1^ every minute (0.41 km·h^−1^) keeping the slope gradient constant during the entire test. Given that the IncT aims to evaluate maximal physiological capabilities as $\dot{V}O_{2max}$, it is important to verify that the protocol was designed for lead athletes to reach their maximal capabilities and to avoid a stop to the test induced by muscular fatigue or non-physiological reasons [23,27].

Firstly, the aim of this study was to investigate whether maximal cardiorespiratory variables are reached by TR runners during the IncT by comparing variables obtained with a verification phase performed thereafter [28,29]. A secondary aim was to evaluate if ventilatory thresholds (VT1 and VT2, respectively) can be determined during the incremental protocol using the IncT.

## 2. Materials and Methods

### 2.1. Participants

Twenty well-trained male TR runners (training volume >8 h per week) were included in the present study (age: 30.1 ± 8.4 years; height: 179.7 ± 6.5 cm; body mass: 68.7 ± 6.6 kg). Their training volume was 7.5 ± 2.1 h per week for 8.8 ± 6.7 years. The participants were healthy, without injuries in the previous 6 months, and were not taking any medication. All subjects did not undergo any strenuous exercise in the three days before the test. Written informed consent was obtained from the subjects, and the study was conducted according to the principles laid down in the Declaration of Helsinski. This study has been approved by “COMITE DE PROTECTION DES PERSONNES SUD MEDITERRANEE IV (ID-RCB: 2019-A03012-55).

### 2.2. Experimental Design

All the participants performed the IncT test on a large and motorized treadmill (S 1930, HEF Techmachine, Andrezieux-Boutheon, France) with a 25% slope. The AS at the beginning of the test was set at 500 m·h^−1^ and increased by 100 m·h^−1^ every minute until exhaustion as described previously. Maximal AS (AS_max_) was considered as the speed in the last stage completed by the athlete. After 30 min of passive recovery, the participants performed a second bout of exercise at a supramaximal intensity to verify the maximal values obtained during the IncT test [24,28,30,31]. In this second bout of exercise, athletes started to run on the treadmill set at the same gradient slope (25°) for a 5-min warm-up at a speed set at 60% of the AS_max_. After this phase, athletes ran at 105% of the AS_max_ with the objective of maintaining the speed until exhaustion (Tlim). The experimental design can be observed in Figure 1.

During both situations, respiratory gas exchanges and heart rate were collected breath by breath during 2 min at rest and during the exercise by a Metalyzer 3B-R3 system (Cortex Biophysics, Leipzig, Germany). The breathing flow was measured through a bi-directional digital turbine and a 220-cm sample line tube collected inspired and expired air to measure O_2_ and CO_2_ concentrations. All data were transmitted wirelessly to a computer using MetaSoft Studio^©^ studio software 5.5.2 (Cortex Biophysics, Leipzig, Germany) to calculate the O_2_ consumption ($\dot{V}O_{2}$, L·min^−1^) and CO_2_ output ($\dot{V}CO_{2}$, L·min^−1^). Before each test, a flow sensor was calibrated with a 3 L syringe and gas sensors were calibrated with air ambient and reference gas (15% O_2_, 5% CO_2_) as recommended by the manufacturer.

$\dot{V}O_{2max}$ (mL·kg^−1^·min^−1^) was calculated using the highest $\dot{V}O_{2}$ measurement average in 30 s for IncT and Tlim tests. In the same time period, the average of the following physiological variables was processed to obtain maximal values: the respiratory exchange ratio (RER), heart rate ($HR_{max}$) in bpm, minute ventilation ($\dot{V}E_{max}$) in L.min^−1^, and breath frequency (BF) in cycles.min^−1^. VT1 and VT2 were determined using the Wasserman and Beaver methods [30] by three unsighted operators. The mean of the two closest values (out of three) was reported. In addition, the presence of the plateau of $\dot{V}O_{2}$ was investigated and confirmed when the $\dot{V}O_{2}$ remained stable (less than 150 mL O_2_ variation) during at least 30 s despite the workload increase [31].

For both tests, blood lactate concentration $\left[La\right]\;$ was measured 1, 3, 5, and 7 min after the end of exercise and the highest value was retained. A capillary blood sample was extracted from the finger (0.5 μL) and analysed using the Lactate Scout+ (EKF Diagnostic, Cardiff, United Kingdom). At the end of each step, the subjective perception (RPE) of effort was quoted by each participant using the rating of perceived exertion scale from 0 to 10.

### 2.3. Data and Statistical Analyses

Results are presented as means ± standard deviation (SD). For both IncT and Tlim, the data were average over 15 s [28]. For all participants, criteria to conclude that maximal data were reached during IncT were a combination of the following variables: (i) plateau of $\dot{V}O_{2}$; (ii) RER ≥ 1.1; (iii) $\left[La\right]$ > 8 mmol·L^−1^ after the test; (iv) the participants stopped the exercise despite strong verbal encouragement; and (v) HR greater than 90% of the predicted maximum (220 bpm−age) [23,28,31]. We calculated the percentage of tests that validated each criterion. The Kolmogorov–Smirnov test was used to investigate data distribution, and it confirmed that all variables were normally distributed. Thus, the Student t test for paired values was performed to compare the maximal data obtained during the IncT and Tlim. For all statistical comparisons, statistical significance was established as *p* < 0.05.

## 3. Results

Results of maximal data during the IncT and Tlim are presented in Table 1. During the IncT test, 16 athletes (80%) reached a plateau of $\dot{V}O_{2}$, 19 athletes (95%) obtained RER values higher than 1.10, and all athletes obtained $\left[La\right]$ over 8.0 mmol·L^−1^ and $HR_{max}$ ≥ 90% of the predicted maximum value. This means that 16 athletes (80%) fulfilled the four pre-determined maximal criteria, 19 fulfilled more than three criteria (95%) and all athletes fulfilled more than two criteria.

Maximal values obtained during the Tlim were not statistically different from maximal data obtained during the IncT (Table 1). VT1 and VT2 were successfully determined by the three operators during the IncT for every subject and the corresponding values are reported in Table 2.

## 4. Discussion

It is well known that the determination of $\dot{V}O_{2max}$ is highly influenced by the mode of testing (i.e., treadmill, cycle ergometer, rowing ergometer, etc.) [28]. For a given locomotion, the characteristics of the test (duration of the step, increment in intensity) can also alter the different determined parameters at submaximal (ventilatory thresholds), and maximal levels. Previous studies already demonstrated that $\dot{V}O_{2max}$ is influenced by the gradient slope used during an incremental running test [17,18,27,32].

As TR is characterized by uphill and downhill sections and triggers specific lower limb adaptations [12,14], it is of great importance to use a tailored test when evaluating TR specialists. In the present study, we investigated whether the IncT, an incremental test based on the AS, satisfies the usual criteria of maximal cardiorespiratory effort and compared data obtained at exhaustion with data obtained during a verification phase consisting in a time trial exercise conducted at 105% of the AS reached during the IncT [28].

The plateau in oxygen uptake is usually considered as the best evidence that the actual $\dot{V}O_{2max}$ is reached during an incremental test [28]. In our study, a plateau was found in 80% of the athletes, which is in accordance with the literature for such a population [28]. However, the usefulness of the plateau criterion to determine $\dot{V}O_{2max}$ has been challenged, not because of the plateau itself, but because of the methodology used in its identification as underlined by Schaun in 2017. Secondary criteria have been judged necessary to be defined when a plateau is not evident. The most common ones are thresholds in the RER; age predicted HRmax; blood lactate concentration. These secondary criteria were fulfilled by all the TR runners except for one for whom the RER was 1.09. Altogether, it means that based on the classically used criteria, $\dot{V}O_{2max}$ was measured in 95% of the TR runners during the Inc based on three or four criteria [23,24,33]. Only one (5%) TR runner fulfilled only two criteria ($\left[La\right]$ and HR_max_).

However, these secondary criteria have also been challenged [28]. For example, these criteria were shown to be unable to differentiate those who demonstrated a plateau in $\dot{V}O_{2}$ from those who did not [23,33]. Thus, the verification phase was recommended to confirm that maximal aerobic capacity was effectively measured during the incremental test [24,28,31]. In our study, we performed a time trial exercise conducted at 105% of the AS 30 min after the IncT. Such characteristics (duration of the period of rest, intensity) are in line with what is commonly used and proposed [28]. When comparing the maximal value measured during the Tlim and IncT, no significant differences were observed (<3%) (Table 1). Noticeably, $\dot{V}O_{2max}$, *VE*_max_, BF, *HR*_max,_ RPE, or $\left[La\right]$ data were similar (Table 1). Hence, the progressive test based on the AS proposed allowed the measuring of maximal cardiorespiratory data in the TR specialists population.

The design of the IncT was based on previous experimentations and studies [18,26,33] to set the slope, starting speed and incrementation, and obtain an adequate duration [33]. Even if the 25% gradient slope looks more appropriate to obtain maximal values [18] and more convenient for treadmill limitation, we noted that a similar protocol could be performed on a steep slope as 40% with similar AS increment (100 m·h^−1^ or 0.27 km·h^−1^ for treadmill speed). We do not recommend different designs other than a slope between 25 and 40% gradient, a starting speed set at 500 m·h^−1^ and incrementation of 100 m·h^−1^ every minute. A too small increment expands the test duration and could trigger a test stop because of lower limb muscular fatigue. Traditionally, the recommendations for a maximal test duration is about 8–12 min even if Midgley and co-workers broke this dogma by reporting that the maximal treadmill test could be up to 26 min [33]. On the contrary, if the workload increase is too rapid, test progressivity could be too aggressive and woud not prepare athletes to perform at their maximal. Additionally, too rapid an incrementation or too high a starting intensity can render the determination of ventilatory thresholds difficult. In our study, the tests lasted 16.0 ± 1.6 min and the VT1 and VT2 were successfully determined in all the athletes. The values corresponded to the literature (VT1 = 60% and VT2 = 88% of $\dot{V}O_{2max}$) for such TR specialists [34]. It highlights that the IncT test offers the possibility to grade smoothly the workload intensity and to accurately characterize useful physiological (HR, *VE*, *BF*, $\left[La\right]$, $\dot{V}O_{2}$) and mechanical (AS) parameters to tailor training intensity or pacing during competition.

Moreover, we noted that the IncT protocol started at a low speed to allow TR specialists to select naturally their motricity between walking or running. During the test, all participants started by walking and operated a transition to running based on individual capabilities [26]. This information, usually called preferred transition speed running, can be another interesting training marker which could be adjusted or optimized based on physiological and biomechanical values recorded during the IncT.

Limitation: In order to standardize as much as possible, this study was performed on a large treadmill instead of in field conditions. In such condition, granularity of the surface and regularity of the slope can differ from the field [35] and required less adjustment from athletes. These elements may require an adjustment to transfer directly the AS obtained in treadmill conditions to the field. In addition, this test was designed to assess trail runner specialists in a situation close to the practice. Running on this gradient slope (25%) is not something normal for other populations such as road runners or untrained athletes. We do not recommend the use of this test on populations other than TR specialists to investigate the physiological limitations or define training intensities.

## 5. Conclusions

The IncT appears suitable and valid for the determination of maximal cardio-respiratory variables during uphill running. Thus, the IncT allows appropriate evaluation of TR specialists’ aerobic fitness in situation closer to practice or competition and lead to the establishment of training markers that can be used by coaches during the training process or during pacing in races. In addition, the IncT was performed on a grade slope (25%) and calibrated by AS to obtain mechanical information directly transferable to the field to calibrate or verify intensity during practice. Based on technology integrated in an HR monitor as a GPS or barometer, the AS can be used in real conditions with an appropriate reference (AS_max_) to adjust or control the mechanical intensity in real practice. To our knowledge, the IncT is the first valid test for trail runners that allows us to assess physiological variables in specific conditions and that outputs mechanical indicators transferable to field conditions. Analogous to any testing process, all variables obtained during the IncT allowed a tracking fitness level or performance potential for trail runners [13,17].

## Figures and Tables

**Figure 1 sports-10-00174-f001:**
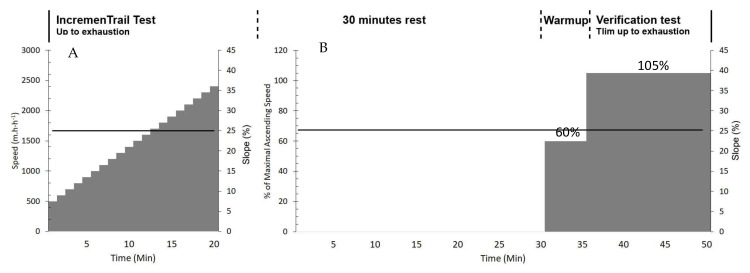
Experimental design description. In the graphic (**A**), the grey area represents the ascending speed in km·h^−1^ where black solid line represents the slope gradient in degrees; in the graphic (**B**), the grey area represents the ascending in percentage of maximum ascending speed obtained during A where black solid line represents the slope gradient in degrees.

**Table 1 sports-10-00174-t001:** Physiological values in IncremenTrail (IncT) and time limits exercise (Tlim) tests (*n* = 20).

	IncT		Tlim		*p* Value
Test duration (s)	943	±92	162	±68 *	0.0012
$\dot{V}O_{2max}$ (mL·kg^−1^·min^−1^)	65.6	±7.6	65.5	±9.0	0.83
Duration of $\dot{V}O_{2max}$ plateau (s)	58.5	±39.2	58.3	±29.9	0.79
$HR_{max}$ (b·min^−1^)	185	±11	183	±10	0.63
$\dot{V}E_{max}$ (L·min^−1^)	170.4	±20.3	171.9	±23.1	0.72
BF (cycles·min^−1^)	55	±7	57	±7	0.68
$\left[La\right]$ (mmol·L^−1^)	17.1	±3.5	15.9	±2.6	0.41
RER	1.12	±0.02	1.13	±0.02	0.55
RPE	8.1	±0.6	8.6	±0.6	0.21

Data are presented as mean ± SD; * Significant difference between IncT and Tlim exercise, *p* < 0.05; $\dot{V}O_{2max}$, maximal oxygen consumption; $HR_{max}$, maximal heart rate; $\dot{V}E_{max}$, maximal minute ventilation; BF, breathing frequency; $\left[La\right]$, blood lactate concentration; RER, respiratory exchange ratio; RPE, rating of perceived exertion.

**Table 2 sports-10-00174-t002:** Physiological values corresponding to the ventilatory thresholds (VT1, VT2) in IncremenTrail test (*n* = 20).

	VT1		VT2	
$\dot{V}O_{2}$ (mL·kg^−1^·min^−1^)	39.8	±8.2	57.4	±7.9
$\dot{V}O_{2}$ (% $\dot{V}O_{2max}$)	60.4	±9.3	87.6	±7.6
HR (b·min^−1^)	143	±19	173	±12
HR (% HR_max_)	77.1	±7.1	93.6	±3.1
AS (m·h^−1^)	1115	±206	1565	±200
AS (% AS_max_)	61.1	±8.7	86.0	±8.0
$\dot{V}E$ (L·min^−1^)	67.1	±14.8	117.0	±19.1
BF (cycles·min^−1^)	26	±6	37	±8

Data are presented as mean ± SD; VT1: first ventilatory threshold, VT2: second ventilatory threshold, $\dot{V}O_{2}$: oxygen consumption volume, HR: heart rate, $\dot{V}E$: minute ventilation, BF: breath frequency, AS: ascending speed.

## Data Availability

Not applicable.

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
