# Peer review of "Specific Incremental Test for Aerobic Fitness in Trail Running: IncremenTrail"

_sports, 2022, doi:10.3390/sports10110174_

Round 1

Reviewer 1 Report

I suggest to reconsider the title as the study seems to be about the validity of an intermittent ascending running protocol.

Please adhere to the Sports template, e.g. there is no running title and author in L16 can be deleted. References style is also not adhered to.

Please be consistent with name of the test as in L19 is different than in the title.

Please clarify with details of the incremental test, i.e. slope, initial speed, and increment speed.

I suggest to refer to doi: 10.1080/15438627.2021.1917405.

L19. I suggest to delete “dedicated”.

L22. “Tim” should be “Tlim”.

L23. Change “respiratory rate exchange ratio” to “respiratory exchange ratio”

I assume the “rate of exercise perception” is the “rate of perceived exertion”.

L27. It is more common to express units as “mL·kg-1·min-1”. Please revise throughout the manuscript.

L29. Please clarify that it is the age-predicted maximum heart rate.

L29. Please change “185 ± 10.5” to “185 ± 11”. Please express heart rate without decimal places throughout the manuscript.

L30. The conclusion is not completed supported by the data in the abstract. Please revise as there is no information on specific intensities.

L46. It reads as if neuromuscular fatigue is unique for downhill and uphill running. I suggest to revise.

L49. Change “steps” to “step”.

L79. Please clarify “AS determine at 25 and 40%”

L94. Please clarify “induce exponential”.

L101. I suggest to simplify “500 m·h-1 of AS (25% slope and 2,06 km·h-1 )”. Just mention.

Throughout the manuscript, I suggest to provide running speed in km/h.

L109. The introduction needs to clarify/rationalise why the interest in determination of the ventilatory thresholds. It is an aim of the study but not been given attention.

L120. Change “maid” to “laid”.

L131. Please provide the gradient of the Tlim test.

L161. How did you define the plateau? Please clarify.

L166. What is meant by “Data collected during IncT test were confronted to criteria of maximal effort”.

Table 1 and 2. I suggest to express breathing frequency without decimal places and change “breath” to “breathing”.

Table 2. Change “ventilator” to “ventilatory”.

Throughout the manuscript, VE should be described as minute ventilation.

Ls 180-186. Please correct.

Table 1. Please define “CV”.

L232. Please delete “extremely”.

L239 and L102. 100 m/h with different km/h. Please revise.

L244. Recently? but you refer to a 2008 study.

L245. Please clarify “…if the increment is too important”.

L253. Please provide support that trail runners pace themselves with ventilatory thresholds.

Figure 1 can be deleted.

Author Response

Reviewer 1

Dear,

Thank you for your comments and suggestion in this article. You can find after our updates and answer. Regarding edition of the document, we did not perform this task, MDPI did the formatting. We also updated the formatting. This study is included in a list of works that we are publishing it may help to read this one to understand our approach.

Physiological Implication of Slope Gradient during Incremental Running Test.

Cassirame J, Godin A, Chamoux M, Doucende G, Mourot L. Int J Environ Res Public Health. 2022 Sep 26;19(19):12210. doi: 10.3390/ijerph191912210.

Specific comments

I suggest to reconsider the title as the study seems to be about the validity of an intermittent ascending running protocol.

We do not understand this point. Test is not intermittent but continuous test with constant gradient slope set at 25%. Maybe some confusion in interpretation can be done in the study which compare the test itself with a verification test to ensure that the test can permit to reach physiological exhaustion and avoid wrong positive test. We also added a figure to summarize the experimental design. It may help.

Please adhere to the Sports template, e.g. there is no running title and author in L16 can be deleted. References style is also not adhered to.

Formatting has been done by MDPI , we reprocessed many points during the review, We corrected the correspondence and removed running title. 

Please be consistent with name of the test as in L19 is different than in the title.

Thank you for that, we also remove hyphenation from the title to avoid splitting the name of the test.

Please clarify with details of the incremental test, i.e. slope, initial speed, and increment speed.

I suggest to refer to doi: 10.1080/15438627.2021.1917405.

We added some information about the protocol to inform reader directly in the abstract the reader

L19. I suggest to delete “dedicated”.

Done

L22. “Tim” should be “Tlim”.

Corrected

L23. Change “respiratory rate exchange ratio” to “respiratory exchange ratio”

Corrected

I assume the “rate of exercise perception” is the “rate of perceived exertion”.

Updated

L27. It is more common to express units as “mL·kg-1·min-1”. Please revise throughout the manuscript.

Yes thank you. We updated in the entire document

L29. Please clarify that it is the age-predicted maximum heart rate.

We refreshed with theoretical maximum HR

L29. Please change “185 ± 10.5” to “185 ± 11”. Please express heart rate without decimal places throughout the manuscript.

Done

L30. The conclusion is not completed supported by the data in the abstract. Please revise as there is no information on specific intensities.

Intensity (AS) can be determined at ventilatory threshold and exhaustion. These values are specific intensities useful for training purposes.

L46. It reads as if neuromuscular fatigue is unique for downhill and uphill running. I suggest to revise.

Yes, all prolonged exercise induce neuromuscular fatigue, We refreshed with more muscular damages That’s more in line with Gioandolini et al. study

L49. Change “steps” to “step”.

done

L79. Please clarify “AS determine at 25 and 40%”

This sentence was wrong? VO2 max do not differ when test is performed at 25 or 40% and AS is relevant indicator of intensity.

L94. Please clarify “induce exponential”.

Energy cost is related to speed and gradient slope. If both speed and slope increase simultaneously, energy cost increase massively. Exponential is not the appropriate word, we changed for Abrupted increase

L101. I suggest to simplify “500 m·h-1 of AS (25% slope and 2,06 km·h-1 )”. Just mention.

Throughout the manuscript, I suggest to provide running speed in km/h.

No, we created this test based on practitioners’ feedback who use ascending speed values as reference for training guidance. This paper is the angular stone of following papers. Is it also in line with elevation challenge and TR practitioner comparison of training session or uphill segment.

L109. The introduction needs to clarify/rationalise why the interest in determination of the ventilatory thresholds. It is an aim of the study but not been given attention.

Maximal capabilities and ventilatory thresholds with intensities and physiological values associated are important markers for training guidance in all endurance sport. We don’t think that is necessary to discuss about utility of these point in here. 

L120. Change “maid” to “laid”.

updated

L131. Please provide the gradient of the Tlim test.

We updated the sentence to add the gradient and add a figure 1 to provide quick view of the design and clarify.

L161. How did you define the plateau? Please clarify.

This part was described latter. We moved these information in the metabolic measurement section.

L166. What is meant by “Data collected during IncT test were confronted to criteria of maximal effort”.

This sentence was removed. The idea was the same than previous one.

Table 1 and 2. I suggest to express breathing frequency without decimal places and change “breath” to “breathing”.

Updated

Table 2. Change “ventilator” to “ventilatory”.

Updated

Throughout the manuscript, VE should be described as minute ventilation.

Ls 180-186. Please correct.

We amended as ventilation minute

Table 1. Please define “CV”.

There is no CV, it has been added during edition. We removed CV, there is no coefficient of variation

L232. Please delete “extremely”.

Removed

L239 and L102. 100 m/h with different km/h. Please revise.

We maintained AS which is the purpose of the test. Trail runner specialist use m.h-1 and that’s the logic of such test.

L244. Recently? but you refer to a 2008 study.

We agreed that, recently is not appropriated and we removed this word

L245. Please clarify “…if the increment is too important”.

We refreshed with “workload increase is too fast”

L253. Please provide support that trail runners pace themselves with ventilatory thresholds.

For all long distances second ventilatory threshold intensity is performance criteria and limitation. Perform a long exercise at intensity over second ventilatory threshold may lead to metabolite accumulation and lead to early exercise stop.

Figure 1 can be deleted.

Yes fully agree, this one do not provide any additional relevant information.

Reviewer 2 Report

In summary, this study investigated the validity of dedicated test performed in uphill condition and evaluate trail runners' cardiorespiratory parameters. While the authors found that this protocol can be used to evaluate trail runners' cardiorespiratory limitations, this reviewer is not convinced with its clinical or training application. 

Line 22: Tim --> Tlim

Introduction: I suggest line 96-106 to be moved to Methods section.

Materials and methods: Line 129 - just to make sure, the second test is also done at the same gradient? Please try to describe more on Tlim. 

Line 134 - what does this line represent?

Results/Discussion

-Based on the average VO2 max value, it seems that participants in this study are well-fit individuals/athletes. How would it differ for those who are not trained? Would novice trail runners be able to tolerate this protocol without getting muscular fatigue? If this is the limitation, make sure to discuss them in the limitations.

-Also, I am not convinced with the advantage of this protocol over traditional setting with minimal gradient. Are there data to support that VO2 significantly differ between whether it is tested in the flat surface (or treadmill without gradient or 1-2% gradient) versus the protocol used in the current study?

-The authors briefly mention that trail runners can adopt the parameters obtained from this protocol while training. But given only one gradient (25%) tested, I am not convinced that the parameters can be applied in other gradients and therefore in training as well. Please describe in detail, how the results of this study can have clinical/training utilities or applications. 

Limitation: There are many more limitations than what authors mentioned. First, as written above, generalizability is a big question given that the study seemed to be conducted in highly-trained individuals. The comments I made above can be potential limitations as well. 

Author Response

Dear,

Thank you for your review and comment which help us to improve the quality of the article. Based on your comments, seem to be very hard to convince you if you never practice specific test. The idea behind this test is more or less the same than all specific test: the specificity. Level aerobic test is interesting metrics for level training but is it not useful to set intensity in uphill conditions. Similar approach has been done for all sport specificity with IFT 30-15 or YOYO test for intermittent training purpose for example or specific tennis test (On the Use of a Test to Exhaustion Specific to Tennis (TEST) with Ball Hitting by Elite Players, Brechbuhl et al. 2016)

Regarding the specificity, we previously demonstrated that specific test could permit to reach higher physiological value when TR specialist performed test in gradient slope at 25 or 40% compared to Level or 15%, because of muscular mass involve and vertical force production required.  See Cassirame et al. 2022. Physiological Implication of Slope Gradient during Incremental Running Test. International Journal of Environmental Research and Public Health

We tested more than 500 athletes in last 5 years from various level and difference between practitioner is very important. The relationship between Maximal ascending speed and Performance is clearly stronger than level maximal aerobic test and performance.

You can see below a part of our dataset with various runners (TR specialist and not) with their result at both tests Level and uphill test. VO2 max is the biggest predictor for both but similar VO2 or MAS so not lead to same Max AS.

Line 22: Tim --> Tlim

We updated this one Updated

Introduction: I suggest line 96-106 to be moved to Methods section.

Is it something that we think before, but this test need to be presented before to clearly present the aim of the study before the methodology.

Materials and methods: Line 129 - just to make sure, the second test is also done at the same gradient? Please try to describe more on Tlim. 

Yes, other reviewer made same comment, we improve this part and add a figure 1. to clarify the experimental design

Line 134 - what does this line represent?

It was paragraph title; we removed those one.

Results/Discussion

-Based on the average VO2 max value, it seems that participants in this study are well-fit individuals/athletes. How would it differ for those who are not trained? Would novice trail runners be able to tolerate this protocol without getting muscular fatigue? If this is the limitation, make sure to discuss them in the limitations.

The protocol is designed for trail runner practitioner, not for road runner or other. But I can confirm that if people are not specialist or severely unfit, lake of force will be the first limitation and VO2 max will not be reach. We will publish latter other study on performance criteria of the test and database with larger range of fitness level. For all TR athlete with more than 1 year of training in the discipline, we never reach lower VO2max during Inct than Level.

-Also, I am not convinced with the advantage of this protocol over traditional setting with minimal gradient. Are there data to support that VO2 significantly differ between whether it is tested in the flat surface (or treadmill without gradient or 1-2% gradient) versus the protocol used in the current study?

One more time see See Cassirame et al. 2022. Physiological Implication of Slope Gradient during Incremental Running Test. International Journal of Environmental Research and Public Health published last month and some others

Kasch, F.W.; Wallace, J.P.; Huhn, R.R.; Krogh, L.A.; Hurl, P.M. VO2max during Horizontal and Inclined Treadmill Running. J. Appl. Physiol. 1976, 40, 982–983. https://doi.org/10.1152/jappl.1976.40.6.982.

Pokan, R.; Schwaberger, G.; Hofmann, P.; Eber, B.; Toplak, H.; Gasser, R.; Fruhwald, F.M.; Pessenhofer, H.; Klein, W. Effects of

Treadmill Exercise Protocol with Constant and Ascending Grade on Levelling-off O2 Uptake and VO2 Max. Int. J. Sports Med. 1995, 16, 238–242. https://doi.org/10.1055/s-2007-972998.

Scheer, V.; Ramme, K.; Reinsberger, C.; Heitkamp, H.-C. VO2max Testing in Trail Runners: Is There a Specific Exercise Test Protocol? Int. J. Sports Med. 2018, 39, 456–461. https://doi.org/10.1055/a-0577-4851

In addition, it is not only a question of VO2, is it also about specific locomotion of uphill which is corresponding to the practice. Between 20 and 40% AS is more accurate mechanical intensity marker than horizontal speed.

-The authors briefly mention that trail runners can adopt the parameters obtained from this protocol while training. But given only one gradient (25%) tested, I am not convinced that the parameters can be applied in other gradients and therefore in training as well. Please describe in detail, how the results of this study can have clinical/training utilities or applications. 

As you can see in Cassirame et al. 2022. Physiological Implication of Slope Gradient during Incremental Running Test. International Journal of Environmental Research and Public Health published last month and some others. VO2, HR and ascending speed obtain between 25 and 40 % are similar. Trail running is mainly performed is such slopes or steeper

Limitation: There are many more limitations than what authors mentioned. First, as written above, generalizability is a big question given that the study seemed to be conducted in highly trained individuals. The comments I made above can be potential limitations as well. 

We added a comment in the limitation section to recommend utilization of this test only for TR specialist.

Round 2

Reviewer 1 Report

In abstract, please change “2,06” to “2.06” and in L101.

Please use in the manuscript the reference doi: 10.1080/15438627.2021.1917405. This is a manuscript on a topic related to your study.

I suggest to use bpm (breathes per minute) as unit for breathing frequency.

L23. I suggest to change “Maximal oxygen consumption” to “Oxygen consumption”.

L79. Please revise “and ?̇?2??? determine at 25 and 40% 79 do not differ at ventilatory thresholds or exhaustion” as at ventilatory thresholds there is not maximum oxygen consumption.

L154. Change “ventilation minute” to “minute ventilation” throughout the manuscript.

L168. Change “pparticipants” to “participants”.

Tables 1 and 2. What are the values between () after the SDs. Please clarify.

Author Response

Dear,

Thank you for your relevant comment. Your suggestion was really appreciated.

We updated the document based one your suggestion.

See below for specific answers.

In abstract, please change “2,06” to “2.06” and in L101.

Updated

Please use in the manuscript the reference doi: 10.1080/15438627.2021.1917405. This is a manuscript on a topic related to your study.

Yes sure, this one was published after that we started to write our own. I inserted this reference in the introduction.

I suggest to use bpm (breathes per minute) as unit for breathing frequency.

We previously received contradictory comment to avoid confusion with HR. Previous paper published by our team used Cycle/min. We rather keep this unit for consistence.

L23. I suggest to change “Maximal oxygen consumption” to “Oxygen consumption”.

Updated

L79. Please revise “and ?̇?2??? determine at 25 and 40% 79 do not differ at ventilatory thresholds or exhaustion” as at ventilatory thresholds there is not maximum oxygen consumption.

Yes, true. We refreshed the sentence.

L154. Change “ventilation minute” to “minute ventilation” throughout the manuscript.

Updated

L168. Change “pparticipants” to “participants”.

Corrected

Tables 1 and 2. What are the values between () after the SDs. Please clarify.

We removed these numbers, not used in this study.

Reviewer 2 Report

The authors addressed the concerns I had. 

Author Response

Thank you